# The Functional Heterogeneity of Neutrophil-Derived Extracellular Vesicles Reflects the Status of the Parent Cell

**DOI:** 10.3390/cells9122718

**Published:** 2020-12-18

**Authors:** Ferenc Kolonics, Viktória Szeifert, Csaba I. Timár, Erzsébet Ligeti, Ákos M. Lőrincz

**Affiliations:** 1Department of Physiology, Semmelweis University, 1085 Budapest, Hungary; kolonics.ferenc@med.semmelweis-univ.hu (F.K.); szeifert.viktoria@med.semmelweis-univ.hu (V.S.); timar.csaba@med.semmelweis-univ.hu (C.I.T.); 2Second Department of Internal Medicine, Szent György Hospital, 8000 Székesfehérvár, Hungary

**Keywords:** extracellular vesicles, microvesicles, neutrophils, anti-inflammatory effects, pro-inflammatory effects, inflammation, antibacterial effect, intercellular communication

## Abstract

Similar to other cell types, neutrophilic granulocytes also release extracellular vesicles (EVs), mainly medium-sized microvesicles/microparticles. According to published data, authors have reached a consensus on the physical parameters (size, density) and chemical composition (surface proteins, proteomics) of neutrophil-derived EVs. In contrast, there is large diversity and even controversy in the reported functional properties. Part of the discrepancy may be ascribed to differences in the viability of the starting cells, in eliciting factors, in separation techniques and in storage conditions. However, the most recent data from our laboratory prove that the same population of neutrophils is able to generate EVs with different functional properties, transmitting pro-inflammatory or anti-inflammatory effects on neighboring cells. Previously we have shown that Mac-1 integrin is a key factor that switches anti-inflammatory EV generation into pro-inflammatory and antibacterial EV production. This paper reviews current knowledge on the functional alterations initiated by neutrophil-derived EVs, listing their effects according to the triggering agents and target cells. We summarize the presence of neutrophil-derived EVs in pathological processes and their perspectives in diagnostics and therapy. Finally, the functional heterogeneity of differently triggered EVs indicates that neutrophils are capable of producing a broad spectrum of EVs, depending on the environmental conditions prevailing at the time of EV genesis.

## 1. Introduction

Extracellular vesicles (EVs) are heterogeneous, phospholipid bilayer-bordered subcellular structures secreted by both pro- and eukaryotic cells spontaneously, upon stimulation or during apoptosis [1]. Since their first identification as cellular debris [2], we learned that they play an active role in intercellular communication. EVs carry biologically active molecules such as nucleic acids (DNA, RNA, miRNA, etc.), proteins, carbohydrates and lipids. Apart from their specific cargo, they possess common proteins and lipids derived from the endosomes, ER, cytosol, or plasma membrane of the parent cell. All of these components inside or on the surface and their combined pattern result in the final complex that exerts the biological function(s). The sorting mechanism that is behind the cargo selection is a well-regulated process reviewed earlier [3]. Regarding the morphology, EVs are spherical structures in most cases, but their shape is very variable according to environmental circumstances [4,5]. Due to their large surface to volume ratio, they are highly efficient for surface interactions with cells and extracellular molecules. Since EVs are heterogeneous in size and biogenesis, the most often used classification is based on these aspects. Exosomes are the smallest EV type (ca. 30–100 nm), produced via the endosomal network and released upon fusion of multivesicular bodies with the plasma membrane. The exosome generation may be the result of both endosomal sorting complexes required for transport machinery (ESCRT)-dependent, and ESCRT-independent processes. Exosomes are enriched in certain endosome markers, such as CD63, CD9 and CD81 tetraspanins [6]. The microvesicles (also called microparticles or ectosomes) are medium-sized vesicles that vary between 100 and 1000 nm in size and are formed by budding from the plasma membrane. The release of the medium-sized EVs is associated most likely with the change of the membrane asymmetry as phosphatidylserine (PS) is exposed in the outer leaflet. This PS exposition is a result of a calcium-dependent activation of scramblase, floppase and the inhibition of flippase [7,8,9]. The apoptotic bodies are released similar to microvesicles by cells undergoing apoptosis, and their size may exceed 1000 nm, but not necessarily. Apoptotic vesicles may contain DNA or histones as specific markers of cell death [4,10].

The field of EV research is growing rapidly, so there are a large number of excellent reviews that summarize in a systematic way the current knowledge on physical and chemical characteristics, biogenesis and composition of different types of EVs, as well as the methodology to investigate them [11,12,13,14,15,16,17]. The aim of this review is to summarize the current knowledge on neutrophilic granulocyte (polymorphonuclear cell, PMN)-derived EVs. By demonstrating the functional heterogeneity of PMN-EVs we try to replace the binary (“to be, or not to be”) concept of EV production by a new “continuous spectrum” concept, where the released EVs reflect the state of the cell of origin. To do this, first we present some examples from the EV biology in general to indicate the significance of EVs and to put PMN-EVs in context.

### 1.1. EVs in Intercellular Communication

EV production is an independent method of intercellular communication, next to humoral and cell-to-cell contact signal transmission. The EVs are potent carriers of biologically active molecules, and apart from the maintenance of homeostasis, they can also influence various pathological functions both of the recipient and the parent cells [1,18]. They are involved in antigen presentation as antigen-presenting cells are able to release EVs containing peptide-MHC I or II complexes [19,20,21] and involved in cell-to-cell transfer of receptors [22] or RNA [23,24] thereby influencing or reprogramming neighboring cells and often promoting tumorigenesis [22,25]. Extracellular vesicles are also involved in a plethora of immunological signal transmissions that have been reviewed previously [21,26,27,28]. Almost every kind of leukocyte-derived EVs are reported to influence the function of other cells. For example, monocyte-derived EVs increase the secretion of IL-8 and MCP-1 in airway epithelial cells [29], IL-6 and MCP-1 in podocytes [30], and TNF-α and IL-6 in monocytes and macrophages via the autocrine or paracrine way [31]. T-cell-derived EVs initiate the secretion of both pro-inflammatory and anti-inflammatory cytokines in monocytes [32] and activation of mast cells [33], and decrease the NO production in endothelial cells [34]. It is also shown that CD4 co-receptors on exosomes from T-cells reduce HIV-1 infection in vitro [35]. Eosinophil-secreted exosomes may influence the pathogenesis of asthma [36] and even mesenchymal stromal/stem cells (MSC) are able to produce EVs with immunomodulatory effects [37]. However, EVs are used for immune modulation by pathogens as well. Several methods of how pathogens try to compromise the immune system by bacterial EVs [38] or by hijacking exosomes by viruses [39] or by impairing T-cell efficacy [40] have been described.

### 1.2. Non-Cellular Effects of EVs

Besides their effects in intercellular signal transmission, EV characterization has identified biological processes that are related directly to EVs. These EV effects can be observed without the presence of a cell transmitting the effect, usually linked to the specific surface of the EVs and reflecting the functions of the parent cells. The most known non-cellular effects are linked to platelet-derived EVs. Many studies show that platelet EVs promote coagulation and thus play an important role in the maintenance of homeostasis [41]. Their pro-coagulant activity is linked to the PS exposure on the EVs that can provide a catalytic surface for assembly of the coagulation complexes [42]. EVs derived from platelets and erythrocytes are able to initiate thrombin generation in a FXII-dependent manner, while monocyte-derived EVs trigger coagulation through tissue factor (TF) [43]. In contrast, there are studies pointing at the anti-coagulant properties of certain EVs by the support of Protein C, Protein S, TF pathway inhibitor and plasmin function [44,45,46,47,48]. Other non-cellular functions of EVs were observed in defense mechanisms against microorganisms: tracheobronchial epithelial cells produce exosome-like vesicles with antiviral activity [49] and sunflower EVs inhibit fungal growth [50].

### 1.3. EVs in Pathological Conditions

The presence of EVs is also shown in several pathological processes. The most intensively studied area is their role in tumor progression [51,52]. Al-Nedawi et al. were the first to demonstrate the transfer of oncogenic EGFR from glioma tumor cells to naive tissue cells (endothelial cells) by EVs [22]. Now we know that EVs are able to directly stimulate tumor growth and angiogenesis. By the secretion of metalloproteinases, they are also able to promote matrix remodeling. There are publications that describe the ability of melanoma-cell-derived EVs to promote immune escape by modulating T-cell activity [53,54]. It has also been reported that tumor-derived EVs play a role in the differentiation of fibroblasts into cancer-associated fibroblasts, therefore inducing a tumor-promoting stroma. They can induce the differentiation of MSC and other bone marrow-derived cells to become tumor-supportive cells by carrying TGF-β and miRNAs [55,56].

EVs play a role in immunologic processes as well. It has been shown in rheumatoid arthritis that platelet-derived EVs carry immune complexes and activate neutrophils as well as maintain inflammation in the synovial fluid [57]. In systemic lupus erythematosus it was observed that EVs expressed pro-inflammatory immune complexes [58]. In term of sepsis, there are also several data regarding the contribution of the EVs [59]. Platelet-derived EVs of septic patients were shown to induce apoptosis in endothelial cells and cardiomyocytes that lead to vascular dysfunctions and cardiomyopathy [60]. The immunomodulatory effect of EVs is shown in acute myocardial infarction [61], in COPD [62], in inflammatory bowel disease [63], in essential hypertension [64] and in type 2 diabetes mellitus [65]. It has been also demonstrated that EVs can take part in neurological diseases by their association with amyloid-β peptides in the case of Alzheimer’s disease and α-synuclein protein in the case of Parkinson’s disease [66,67].

### 1.4. Diagnostics and Therapy

Since EVs can be detected in every body fluid and could be found in every physiological and pathophysiological process, the diagnostic possibilities in EV research are great. In cancer research, there is a huge potential in identifying tumor-derived EVs as biomarkers through liquid biopsies: EVs may serve for tumor screening, diagnosing, staging and also for the prediction of therapy outcome [24,68]. Apart from cancer diagnostics, there are a wide range of application possibilities of EVs e.g., in liquid biopsy of hypertensive disorders in pregnancy, in organ transplantation, and in cardiovascular diseases [69]. In addition, the European Society of Cardiology published a position paper for the analysis and translational application of EVs focusing on the diagnosis and therapy of the ischemic heart disease [70]. Recently there have already been some examples for commercial development of EV diagnostic tools such as the ExoDx Prostate^®^ (Exosome Diagnostics, Waltham, MA, USA) urine exosome gene expression assay, designed for assisting the decision regarding the necessity of needle biopsy [71].

Several research groups work on therapeutic utilization of EVs. The most promising results come from mesenchymal stem cell derived EVs that are transiting rapidly towards clinical applications. Detailed reviews have summarized the current knowledge in this field [72,73,74] demonstrating important observations, for instance in repairing the damaged tissues in myocardial lesions [75,76,77,78] or in joint diseases by their anti-inflammatory properties [79]. Moreover, there are several promising works on the application of the EVs as targeted drug delivery systems. Engineering EV production can help targeted therapy, since it makes it possible to increase the EV’s binding specificity to cancer cells [72,80]. Although there are many challenges and questions in the field of EV-based therapy (e.g., biodistribution, EV clearance, handling and storage of therapeutic EV samples), the potential benefits could be great and will hopefully be achieved in the near future [81].

As can be seen from the brief overview of EV research above, there is plenty of information on the biological effects of EVs derived from several cells, and some publications also investigate the role of these EVs in pathological conditions and their diagnostic and therapeutic potential. Like other cells, neutrophilic granulocytes (the most abundant nucleated cells in the peripheral blood) are able to generate EVs. Neutrophils, as central cells of natural immunity, play a prominent role in immunological processes. The investigation of the nature and biological effects of EVs formed from them began at an early stage of EV research and dates back more than 20 years. The knowledge accumulated over the last 20 years about neutrophil EVs brings us to demonstrate how EVs with different functions are produced depending on the stimuli acting on the neutrophil.

## 2. Neutrophil-Derived EVs

Neutrophilic granulocytes belong to the first line defense of innate immune reactions against bacteria and fungi [82,83]. PMN actively communicate with surrounding cells via humoral mediators and surface molecules [84]. It is not surprising that, similarly to other cells, neutrophils are able to secrete extracellular vesicles which affect the function of their biological environment (e.g., other PMNs, macrophages, endothelial cells, vascular and bronchial smooth muscle cells, or the coagulation cascade). In the past two decades many studies have investigated the effects of PMN-EVs in the regulation of the local and systemic inflammatory environment. Intriguingly, the findings are diverse and sometimes even contradictory. This inconsistency could arise from differences in the quality of the PMN isolates, the stimulus used for PMN-EV production, EV isolation procedures, the storage of EV samples and the experimental environment of the investigated target cells. Table 1 summarizes these works.

Overviewing the data in Table 1, it is striking (indicated by question marks) that most methodological descriptions do not mention the purity or the cell composition of the tested sample. It is also obvious that EVs were isolated with DC (differential centrifugation) with or without filtration in almost all studies. This corresponds with the results of the 2015 ISEV survey [139]. There is no consensus on the used centrifugation forces. Typically, there is a short lower-speed centrifugation to sediment cells and debris. This is followed by a longer higher-speed centrifugation to sediment EVs. Although there are notable differences between applied centrifugation parameters, the size distribution and electron microscopic appearance of the isolated EVs are surprisingly similar. Following MISEV 2018 guidelines, our group compared the characteristics of PMN-EV isolates prepared with DC and SEC (size exclusion chromatography). The SEC isolates showed similar physical and functional characteristics to the DC-prepared EV isolates, but the SEC-EV yield was significantly lower [140]. To avoid misinterpretation of results, careful quality control of the parent cells and the isolated EVs is needed. Since neutrophils are short-living and sensitive cells, and peripheral blood already contains heterogeneous PMN populations, control of the cell viability is needed to determine the percent of apoptotic cells [10]. Another aspect of the quality maintenance is the careful handling of EVs after isolation. We have shown that the number, physical characteristics, and biological effects of PMN-EVs are significantly affected even after 24 h storage at 4 °C and −20 °C [141]. Unfortunately, only a few articles provide data on storage conditions (Table 1).

In order to obtain an appropriate quality of a PMN-EV sample (based on our own experience and the facts summarized in Table 1) we recommend: (1) Controlling the viability of EV-producing cells and indicating the percentage of apoptotic cells. This is especially important if you start from buffy coats or from cell cultures. In case of cell culture preparations, it is necessary to remove all EVs originating from the used culture serum to avoid contamination of the EV isolate. (2) In the first centrifugation step, it is recommended to check the amount of EV-like structure in the sedimented debris. (3) Compare the EV isolate obtained with DC preparation to EV isolates obtained by SEC or other intermediate separation protocol [142]. (4) Functional testing should preferably be performed with freshly prepared EVs. If necessary, it is recommended to store EVs at −80 °C [141]. (5) As there is no flawless preparation method, it is recommended to detail the used preparation protocol and the performed controls in the methods, following the MISEV 2018 guidelines [142].

### 2.1. Characteristics of Neutrophil-Derived EVs

Almost every study analyzed the size of extracellular vesicles (Table 1). There is a consensus that the vast majority of PMN-derived EVs belong to medium-sized vesicles (called microvesicles or ectosomes) [95,118,124,131]. The size of the EV spreads mostly between 100 and 700 nm with a modus around 200–300 nm and there was no typical difference observed between differently triggered EVs. This range was confirmed by different examination modalities (Figure 1).

Their appearance on electron micrographs was heterogeneous in size, density and structural content both with conventional TEM [140] and cryo-TEM imaging [122].

There is also consensus that due to their larger size, the greater part of the neutrophil-derived EVs could be analyzed by flow cytometry. PMN-EVs carry typically CD66b, CD11b, CD18 and MPO (myeloperoxidase) on their surface and the greater part could also be labeled with annexinV due to PS exposure [124,143].

Another widely tested parameter is the amount of produced EVs. Due to the various limitations of different detection techniques it is almost impossible to enumerate EVs exactly. Both single particle enumeration methods (nanoparticle tracking analysis, flow cytometry, tunable resistive pulse sensing) and bulk measurements (protein and lipid quantification) should be interpreted with caution due to potential methodological pitfalls. It is advised to use different methods in parallel for better estimation of the EV number. Although we cannot determine the exact number of EVs, we can compare EV populations to a reference population (e.g., to spontaneously produced EV population) to define increased ratios upon stimuli. Here we present a comparison of the most often used stimuli that trigger EV production under comparable circumstances (Figure 2). Data are from [10,124,140]. It is worth noting that many single receptor activators did not increase significantly the EV generation compared to spontaneous EV production. The strongest EV productions were detected when PMNs were stimulated with the natural enemies, with the opsonized particles, or were left to go in apoptosis. Both cases are highly possible fates for the neutrophil.

The protein composition was analyzed in several studies by proteomics: a greater part of the protein content was from the cytoskeleton, the granules and the mitochondria or were signal proteins [10,104,122,124]. Since basic physical and chemical characteristics of PMN-derived EVs triggered by different stimuli do not vary too much (even apoptotic EVs share many common properties with specifically triggered EVs, Table 1), we review here previous studies on PMN-derived EVs according to the used stimuli and the functional heterogeneity of generated EVs.

### 2.2. Neutrophil-Derived EVs in Intercellular Communication

#### 2.2.1. Effect of PMN-EVs Released without Stimulation

PMNs release EVs constitutively and spontaneously without activation (sEV). The production of sEVs is not affected by inhibitors or genetic deficiencies of receptors and signaling molecules [124,125,140]. It is reported that sEVs exert anti-inflammatory effects on *Mycobacterium* infected macrophages [85], and in our experimental settings freshly isolated sEVs also showed anti-inflammatory effects by decreasing ROS (reactive oxygen species) production and IL-8 release from other PMNs [86].

When PMNs are left unstimulated for several hours or in the case of pro-apoptotic environments (e.g., UV-B/C radiation), apoptotic vesicles (apoEVs) are released. Apoptotic EVs in our hands had no effects on pro-inflammatory cytokine production but delayed the ROS production of PMN [86]. In accordance with our results, others found neither pro-inflammatory nor direct anti-inflammatory effects when human [87] or murine macrophages [91], other PMNs [89] and Th cells [90] were exposed to apoEVs. However, there is one study that reported a clear anti-inflammatory effect of apoEVs: monocytes stimulated with LPS in the presence of apoptotic neutrophils for 18 h elicited an immunosuppressive cytokine response, with enhanced IL-10 and TGF-β production and only minimal TNF-α and IL-1β cytokine production [88].

#### 2.2.2. Effect of PMN-EVs Released upon Stimulation with Bacterial Byproducts (fMLP and LPS)

There are more contradicting data in the literature regarding EVs released upon treatment with bacterial products, such as fMLP or LPS. The final effect of produced EVs strongly depends on the used concentration, type and purity of the endotoxin and on the priming state of the cells. Moreover, different target cells may show different cellular answers after the same EV sample treatment. The bacterial byproduct fMLP-triggered PMN-EVs showed dominant anti-inflammatory effects on leukocytes. Schifferli’s group observed that fMLP-triggered EVs interfered with NF-κB signaling in human monocyte-derived macrophages and increased TGF-β1 release [92,94]. These EVs inhibited the inflammasome activation in murine peritoneal macrophages [95] and the maturation of monocyte-derived dendritic cells as well [93]. Similarly, fMLP-induced EVs inhibited the production of IFN-γ and TNF-α but enhanced the release of TGF-β1 by IL-2/IL-12-activated NK cells [96]. Interestingly, GM-CSF-primed and fMLP-triggered PM-EVs contain LTB_4_ that can activate resting neutrophils and elicit chemotactic activity [112].

In contrast to leukocytes, fMLP-induced PMN-EVs exert a rather pro-inflammatory phenotype when incubated with HUVEC (human umbilical vein endothelial cell). Endothelial cells increased IL-8 and IL-6 release after EV treatment [98,99]. A recent study reported that fMLP-induced EVs promote inflammatory gene expression by delivering miR-155, enhancing NF-κB activation and endothelial activation [100]. Dalli et al. examined the endothelium-attached PMN-derived EV functions in many aspects. If PMN were seeded on HUVEC and stimulated with fMLP, the generated EVs inhibited the adhesion of resting neutrophils to HUVEC [97] but upregulated the pro-inflammatory genes in HUVEC cells. Notably, non-adherent stimulation of PMNs by fMLP resulted in anti-inflammatory EV generation [106]. Taken together, these results suggest that endothelium-attached neutrophils produce EVs to inhibit the adhesion of more PMNs to the endothelium, but on the other hand these EVs have pro-inflammatory effects on endothelial cells. In accordance, Rossaint et al. suggest a very interesting mechanism where fMLP triggers a multistep reciprocal crosstalk via arachidonic acid transporting EV production between platelets and neutrophils. Ultimately, the platelet-produced thromboxane A2 elicits a strong neutrophil activation by inducing the endothelial expression of ICAM-1 [103]. Functional studies confirmed that fMLP-stimulated PMN-EVs significantly increased permeability and decreased the transendothelial electrical resistance of a confluent monolayer of human brain microvascular endothelial cells. These findings indicate not just that PMN-EVs interact with and affect the gene expression of endothelial cells but propose the role of EVs in vascular transmigration of PMNs [101]. Importantly, EVs produced during the late stages of extravasation are deposited on the sub-endothelium and maintain the integrity of the microvascular barrier during leukocyte extravasation [109]. In contrast to this precise control of endothelial permeability, fMLP-stimulated PMN are also shown to deposit EVs onto intestinal epithelial cells during transepithelial migration. These EVs promote the disruption of intercellular adhesion, the enhancement of PMN recruitment and the impediment of wound healing partly due to their surface-bound MPO and pro-inflammatory miR-23a and miR-155 content [102,108,111].

Fewer studies investigated the nature of PMN-EVs released upon LPS stimulation. Interestingly, in contrast to fMLP, LPS triggered pro-inflammatory EVs. Habhab et al. examined splenocytes (predominantly neutrophils)-derived EVs and found pro-inflammatory, pro-coagulant and pro-senescent responses in endothelial cells through redox-sensitive pathways [114]. LPS-triggered exosomes increased equine airway smooth muscle cell proliferation [115] and combined LPS and fMLP activation induced EVs enhanced the phagocytosis and the ROS production of monocytes and resting neutrophils [113].

#### 2.2.3. Effect of PMN-EVs Released upon Stimulation with Endogenous Pro-Inflammatory Mediators

PMN priming or activation is also feasible with different cytokines like TNF-α, IFN-γ, GM-CSF, C5a, PAF and IL-8. PMN-EVs from cells activated by either PAF [97] or IL-8 [96] demonstrated anti-inflammatory potential by decreasing PMN recruitment [97], as well as NK cell [96] activation. C5a-induced EVs were rated as anti-inflammatory vesicles as they inhibited macrophage maturation [94]; however, a recently published work showed an opposite effect where C5a-triggered EVs activated resting neutrophils to produce ROS and induced IL-6 secretion [121].

EVs released from TNF-α-stimulated PMNs exhibit a pro-inflammatory profile: they enhance the pro-inflammatory cytokine production and adhesion molecule expression of HUVEC [137] and contribute to genomic instability, inflammation and the impediment of wound healing in intestinal epithelial cells. These latter functions were similar in the case of the strong inflammatory signal transmitting IFN-γ and GM-CSF-triggered EVs [102]. Kahn et al. described that TNF-α-induced EVs transfer functional kinin B_1_-receptors to human embryonic kidney cells and induce calcium influx after stimulation [119]. In contrast, the group of Perretti reported the anti-inflammatory effect of TNF-α-triggered PMN-EVs on human monocyte-derived macrophages and a macrophage-fibroblast-like synoviocyte co-culture system [118].

#### 2.2.4. Effect of PMN-EVs Released upon Stimulation with Pathogens

Opsonized pathogens represent the strongest natural activating signal to PMN via PRR (pattern recognition receptor) and opsonin receptors (e.g., Mac-1/CR3, FcγRs). Our group showed that PMN-EVs released after stimulation with opsonized zymosan carry a powerful pro-inflammatory potential by enhancing ROS production and IL-8 release of resting PMNs and HUVEC [86]. Another group also found pro-inflammatory effects of bacteria and EV aggregates manifesting in enhanced IL-6, IL-1β production and higher CD86 and HLA-DR expression of macrophages [144]. However, this study emphasized the role of aggregated bacteria in the detected pro-inflammatory effects, as EVs alone did not enhance the cytokine release of macrophages. Alvarez-Jiménez et al. also described a pro-inflammatory profile of PMN-EVs released after *M. tuberculosis* infection of neutrophils [105]. On the contrary, Duarte et al. reported diminished bacterial clearance by human monocyte-derived macrophages after treating them with PMN-EVs released after *M. tuberculosis* infection [85]. The difference in outcome of the latter two publications could be explained by the differences in the MOI (multiplicity of infection) for neutrophil infection, the time allotted for EV release by neutrophils, the duration of infection in macrophages and the different isolation protocols for obtaining EVs [105].

#### 2.2.5. Effect of PMN-EVs Released upon Stimulation with Pharmacological Stimuli

Biological significance of pharmacological stimuli-evoked EVs is difficult to interpret; however, as a clean system they can help to understand the mechanism of EV generation. PMA, a potent pharmacologic activator of PMN, can induce EV production as well (Figure 2). As opposed to the powerful overall activating effect of PMA, these EVs are more anti-inflammatory in nature. When PMN-like PLB-985 cells were exposed to PMA stimulation, the generated EVs inhibited monocyte-derived dendritic cell maturation and promoted Th2 polarization [126]. In another study, PMA-induced PMN-derived EVs decreased IL-1β production, but enhanced CD86 expression of human monocyte-derived macrophages [105]. When Ca^2+^ ionophores were used for stimulation, produced PMN-EVs exhibited pro-inflammatory properties by damaging membrane integrity of HUVEC [128] or increasing endothelial activation, vascular senescence and endothelial oxidative stress [114]. L-NAME, a NOS inhibitor, was also shown to induce PMN-EV production. These EVs demonstrated pro-migratory effects with and without a HUVEC layer, when other PMN were exposed to them [129].

#### 2.2.6. Effect of PMN-EVs Released in Pathophysiological Environments

Several studies have examined the presence and biological effects of PMN-derived EVs in pathological conditions. Sepsis is connected to PMNs in multiple ways, since bacteria are the causative agents in most cases. PMNs are affected both in the initiation and in the effector phase of the disease and cytokine storms characteristic in sepsis can also both originate from and affect PMNs. It was reported already at beginning of this century that activated PMNs enhance production of EVs in patients with sepsis [145]. Our earlier work on septic patients confirmed the increased presence of PMN-EVs in the blood and we revealed their ability to form aggregates with bacteria. This sequestration and immobilization of bacteria could contribute to limitation of microbial growth in the early stages of infection [124]. Kumagai et al. found that in cecal ligation and puncture mice models, the injection of antimicrobial peptide, LL-37, induced PMN-EV production that showed antibacterial potential and protected mice from lethal septic conditions by reducing the bacterial load [132]. Another group reported enhanced phagocytic activity, pro-inflammatory activation and increased HLA-DR expression on monocytes exposed to PMN-EVs released in septic patients [130]. The same group also reported a harmful anti-inflammatory and immunoparalytic effect of peritoneal EVs isolated from cecal ligation and puncture model after injection with thioglycolate [133]. Acute pancreatitis can be accompanied by severe systemic inflammation, hence there are immunological traits related to sepsis. A recent study showed that PMN-EVs associated with neutrophil extracellular traps in an animal model of acute pancreatitis contribute to both local and systemic deterioration of inflammation [135].

Beside sepsis, the presence of PMN-EVs was also reported in other infectious diseases. PMN-EVs isolated from the sputum of cystic fibrosis (CF) and primary ciliary dyskinesia patients also showed pro-inflammatory properties: if administered intratracheally in mice, histopathological analysis showed peribronchial and perivascular leukocyte infiltrates [134]. A later study showed a lung extracellular matrix degradation effect of PMN-EVs by surface-bound neutrophil elastase. The proteolytic effect of EV-associated elastase can lead to COPD as EV-bound enzymes are more resistant to alpha-1-antitrypsin. The possibility was also raised that elastase binds to the EV surface after they have been released [107]. A similar post EV release binding of granule proteins has also been previously observed earlier [143].

Neutrophils are strongly involved in immunologic and rheumatologic disorders, too. An early report found elevated PMN-EV levels in patients with acute and chronic vasculitis [146]. In ANCA (anti-neutrophil cytoplasmic antibodies)-associated vasculitis, elevated levels of circulating DNA and TF-expressing neutrophil-derived EVs were observed in sera from patients with active disease. This phenomenon proposes the role of PMN-EVs for the induction of thrombosis in active ANCA-associated vasculitis [136]. TNF-α-primed PMN release EVs when treated with ANCA, and these EVs enhance the ICAM-1 expression of HUVEC [137]. On the other hand, PMN-EVs released upon TNF-α treatment are anti-inflammatory and their effect is more pronounced if the PMNs are derived from RA patients [118].

The presence of PMN-EVs has been confirmed in metabolic-atherogenic diseases. A high fat diet was shown to increase the PMN-EV concentration in blood, and EVs were found to accumulate in atheroprone regions of the vasculature. Since acetylated LDL (low density lipoprotein)-stimulated PMNs produce EVs with high amounts of pro-inflammatory miR-155 that induces endothelial activation, the authors concluded that PMN-EVs contribute to vascular inflammation and atherogenesis [100]. Hyperglycemia also enhances PMN-EV production with higher amounts of IL-1β, which might represent a pro-inflammatory potential [138]. There is also evidence of the involvement of PMN-EVs in acute coronary syndrome. Martínez et al. described an acute neutrophil-derived EV release after percutaneous coronary intervention in acute coronary syndrome compared with stable patients, likely to be reflective of plaque EV content only in vulnerable lesions [147].

### 2.3. Non-Cellular Effects of Neutrophil-Derived EVs

#### 2.3.1. Effect of PMN-EVs on Hemostasis

PMN-EVs exert effects that do not need the contribution of other cells. Their most well-examined non-cellular effect is the pro-thrombotic function. The fundamental observation reveals that many pathophysiological conditions, including bleeding and thrombotic disorders, are accompanied by elevated levels of PMN-EVs [148]. Since EVs expose high amounts of PS on their surface, and some EVs have also been shown to carry TF, many studies describe pro-coagulant effects of PMN-EVs. Some studies reported pro-coagulant activity of TNF-α-primed and ANCA-activated PMN-derived EVs via the extrinsic pathway by TF expression [136,137]. One other work described pro-coagulant effect via the intrinsic pathway in a cecal ligation and puncture model by binding to factor XII [127]. Indirect pro-coagulant activity was described by enhancing TF expression of endothelial cells [98] leading to a secondary generation of pro-coagulant endothelial EVs [114]. Adherent neutrophils released PAF-expressing PMN-EVs upon endotoxin stimulation that activated the platelets [116]. Moreover, platelets were stimulated by conformational active Mac-1 and PSGL-1 expression on PMN-EVs triggered by PAF and LPS [117]. Recently, we have also reported a strong pro-coagulant effect of apoptotic PMN-EVs with or without tissue factor presence. Spontaneously released EVs showed a weaker but still significant pro-thrombotic effect, while EVs from opsonized zymosan activated cells had no effect on coagulation [86].

#### 2.3.2. Anti-Pathogenic Effect of PMN EVs

Our group demonstrated a unique non-cellular effect of a subset of PMN-EVs: vesicles released upon stimulation of Mac-1 receptor with either opsonized pathogens (bacteria or zymosan particles) or selective Mac-1 ligands were capable of forming large aggregates with bacteria. We found these aggregates to inhibit the bacterial growth independently from opsonization of bacteria [124,125,140,149]. Although fMLP-induced PMN-EVs were able to bind bacteria via clusters of CR1 [110], no other receptor activation than Mac-1 resulted in antibacterial EV generation (Figure 3) despite their ability to increase EV production related to spontaneous EV formation (Figure 2). The aggregate forming ability of bacterial infection-induced PMN-EVs were later confirmed in septic patients [131] and in osteomyelitis patients [150]. Similar to bacteria-induced antibacterial EV formation, Shopova et al. described antifungal PMN-EV release upon stimulation with opsonized *A. fumigatus* [122]. Another group showed that infected wild-type (but not CF) murine PMNs release sphingosine, most likely in EVs, which kills *P. aeruginosa* [123].

### 2.4. Role of PMN-EVs in Diagnostics and Therapy

PMNs and EVs play central roles in several infectious and inflammatory diseases; hence there is a great interest in diagnostic and therapeutic options involving PMN-EVs. However, at this point there are only few studies that describe techniques with potential short-term clinical benefits.

A plethora of studies report elevated level of PMN-EVs in septic blood samples [120,124,145,151,152,153,154] and among them we showed at first that they form aggregates with bacteria [124]. Based on our observation, a point-of-care microfluidic chip was proposed, which detects the aggregation potency of EVs isolated from a patient’s serum. The EV-bacteria aggregates were characteristic for bacterial infections but were not present in non-infectious inflammation [131]. Similar results were reported in a rat osteomyelitis model and in patients with osteomyelitis, suggesting a selective aggregating ability of EVs with bacteria which were used to induce their production; however EVs showed some cross-reactivity with other bacteria as well [150]. It is also observed that alpha-2-macroglobulin positive EVs were present in higher amounts in survivors of pneumonia-related sepsis than non-survivors. Thus, alpha-2-macroglobulin expression was suggested as a potential prognostic marker in sepsis [120]. Stiel et al. demonstrated that the PMN-EV/neutrophil ratio, a surrogate of neutrophil activation, correlates with the presence of “disseminated intravascular coagulation” syndrome in septic patients and could serve as another prognostic parameter [155]. Nadkarni et al. propose the use of PMN-EV level measurements to monitor the clinical status of polymyalgia rheumatica patients [156]. Giumaraes et al. made a potentially useful diagnostic observation in infective endocarditis: PMN-EV numbers are higher in blood samples of infective endocarditis patients compared to other bacterial infections. Only PMN-derived EVs were found to be significantly elevated 2 weeks after hospital admission. PMN-EV levels were also significantly higher in non-survivors and were an independent predictor of mortality [157]. PMN-EVs could have pre-event prognostic potential in familial hypercholesterolemia, since patients with higher basal PMN-EV numbers had elevated risk for future major ischemic events [158].

The therapeutic potential of cell-derived EVs in general are reviewed several times [159,160], and only MSC- and DC-derived EVs are likely to be used as therapeutics in the near future. At this time, there exists no test that utilizes the natural therapeutic potential of PMN-EVs, although both the pro-resolving phenotype of PMN-EVs [104,132,133] and the non-cellular anti-pathogenic effect [122,124,131] of PMN EVs suggest a clinical benefit in certain diseases. Importantly, there is a group working on a new drug delivery system using PMN-derived EVs formed by nitrogen cavitation. These nanovesicles are similar to naturally secreted EVs but contain fewer subcellular organelles and nucleic acids. These EVs were loaded with the anti-inflammatory drug piceatannol and dramatically alleviated acute lung inflammation and sepsis induced by lipopolysaccharide [161,162]. Resolvin D2 was also loaded in PMN-EVs to enhance resolution of inflammation in a mouse stroke model system, therefore protecting the brain from damage during ischemic stroke [163].

## 3. Discussion

Extracellular vesicles have become a scientific hot topic in the last decade. Beside other cells, neutrophil-derived EVs have also been intensively examined. The partial explanation of the seemingly controversial effects of PMN-EVs reviewed above may be hidden in the details of cell activation and handling. A vast majority of studies were executed under different circumstances with varying isolation, handling, storing and testing protocols (Table 1). This makes it difficult to compare or combine the results of different studies. Driven by the recognition of these issues also in other fields of EV research, the International Society for Extracellular Vesicles (ISEV) proposed Minimal Information for Studies of Extracellular Vesicles (MISEV) guidelines to help overcome these issues [142,164]. These guidelines and other suggestions originating from milestone observations on handling [165,166,167,168], storing [141,169,170] and controlling the quality of EVs [171,172] should be followed in PMN-EV research as well.

In our recent study, we demonstrated that EVs derived from the same PMNs can have diverse and sometimes even opposing effects depending on the activation signal affecting the parent cells. Consequently, we have concluded that the function of the secreted EVs reflects the activation state of the parent cells [86]. Similar observations were made by Dalli et al. [104]. As we have shown in this review, neutrophils are able to affect all cells relevant to their environment by EV production. This indicates that EV-mediated signaling is divergent not only in terms of the message of signaling, but also in terms of the targeted cells (Figure 4). EVs derived from resting or apoptotic PMNs tend to send anti-inflammatory signal to surrounding cells (Figure 4, group 1–2), which may play an important local role in inhibiting autoimmune processes and in the resolution of inflammation [173,174,175]. The clinical manifestation of this may be the participation of the pathological death of neutrophils in the pathomechanism of autoimmune diseases [176]. On the other hand, in the case of infection, the stepwise activating neutrophil tends to transmit pro-inflammatory EV signals to the neighboring cells to facilitate the current task: diapedesis through a vessel by changing the activation state of the endothelial cells (Figure 4, group 3a,b), recruiting (Figure 4, group 3c) and activating (Figure 4, group 3d) other cells. Finally, upon encountering the natural enemy (the opsonized pathogen), a neutrophil becomes fully activated and secretes EVs with direct antibacterial activity and strong pro-inflammatory effects (Figure 4, group 3d). Neutrophils appear to exhibit remarkable care when producing pro-inflammatory EVs. Single activation with fMLP or TNF-α does not result in a strong pro-inflammatory EV production (Table 2). These EVs stimulate the endothelial cells only by making them capable to anchor immune cells, but at the same time these EVs inhibit the activation of other leukocytes. However, when a second activation signal appears (LPS, GM-CSF) or products of the complement activation are present (C5a, C3bi), neutrophils change to produce EVs with a clear pro-inflammatory effect (Table 2). Apparently, similar to the activation process of lymphocytes, neutrophils also wait for a second validation signal from other immune cells or from the complement system before starting the inflammatory response.

The reviewed spatiotemporal pluralistic function of PMN-EVs suggests that neutrophils are capable of producing a continuous spectrum of EVs, starting from anti-inflammatory EVs up to the pro-inflammatory or even antibacterial EVs, and the properties of the currently secreted EVs reflect the prevailing state of the cell. Thus, we hypothesize a new intercellular signal transmission model, where EV production plays a similar role to cytokine secretion: there are pro-inflammatory and anti-inflammatory EVs similar to the pro-inflammatory and anti-inflammatory cytokines. Moreover, the same EV population can exert different effects on different target cells, since EVs transfer more complex signals due to their more complex structure. Taken together, EV production is a parallel modality of intercellular communication that is complementary to humoral factors and cell–cell contacts in the regulation of immune response.

## Figures and Tables

**Figure 1 cells-09-02718-f001:**
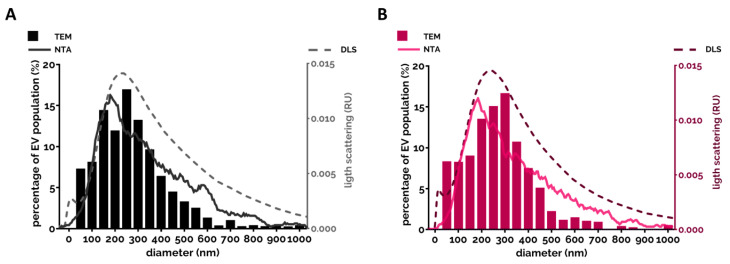
Size distribution of the PMN-EVs. (**A**) Representative size distribution measurement of the spontaneous EV (sEV) population produced by the PMNs after 20 min incubation at 37 °C. EVs were analyzed immediately after isolation. Filled bars represent the transmission electron microscopy (TEM)-based quantification of the size (see [140]). The broken line represents the spectra measured by DLS (see [140]). The grey line represents the NTA measurement (detailed in [86]). (**B**) Representative results of the three different methods on opsonized zymosan-induced EVs.

**Figure 2 cells-09-02718-f002:**
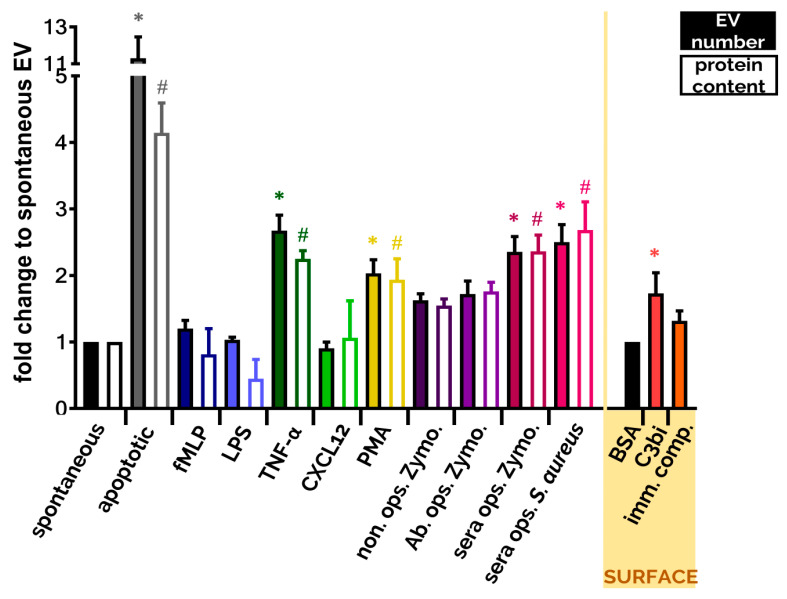
Comparison of EV production of PMNs after different soluble and surface-bound activators. EVs were analyzed immediately after isolation. Isolation and quantification of EVs is detailed in [10,124,140]. Filled bars represent the EV quantification by flow cytometry, empty bars represent the quantification based on protein amount measurement. Cells were treated with the indicated stimuli for 20 min at 37 °C. Stimuli were applied in a final concentration of 1 µM (fMLP), 100 ng/mL (LPS), 20 ng/mL (TNF-α), 100 ng/mL (CXCL12), 100 nM (PMA), 5 µg/mL (Zymosan) or 108/mL *S. aureus*. Error bars represent mean + S.E.M. Data were compared by using one-way ANOVA coupled with Dunett’s post hoc test, *n* = 3 (fMLP, LPS, CXCL12, C3bi surface), 4 (apoptotic), 9 (TNF-α, immune complex surface), 12 bovine serum albumin (BSA), 22 (non ops. Zymosan, Ab. ops. Zymo., sera ops. Zymo.), 32 (sera ops. *S. aureus*), 40 (sEV, PMA). On the “SURFACE” panel we show the FC quantification of EV production of adherent PMN on BSA surface (20 µg/mL), on C3bi surface (50 µg/mL) or on immune complex surface (imm. comp., 20 µg/mL). Error bars represent mean + S.E.M. Data were also compared by using one-way ANOVA coupled with Dunett’s post hoc test. Level of significance (*p* < 0.05) is indicated by * for EV numbers and by # for protein amount.

**Figure 3 cells-09-02718-f003:**
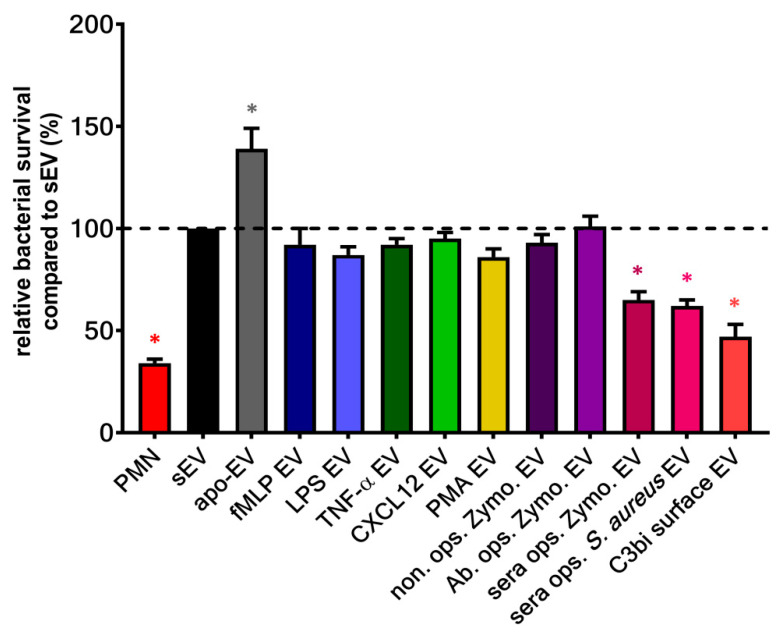
Bacterial survival in the presence of different types of neutrophil EVs. EVs were tested immediately after isolation. The amount of the applied EVs was normalized to protein content. The optical-density-based quantification of the bacterial survival of *S. aureus* was carried out as detailed in [149]. Error bars represent mean + S.E.M. Data were compared to sEV by using one-way ANOVA coupled with Dunett’s post hoc test; *n* = 3 (PMA EV), 4 (fMLP EV, TNF-α EV, LPS EV, CXCL12 EV, C3bi surface EV, apo EV), 11 (non ops. Zymo. EV, Ab. ops. Zymo. EV, sera ops. Zymo. EV), 25 (PMN, sEV, sera ops. *S. aureus* EV). Level of significance (*p* < 0.05) is indicated by *.

**Figure 4 cells-09-02718-f004:**
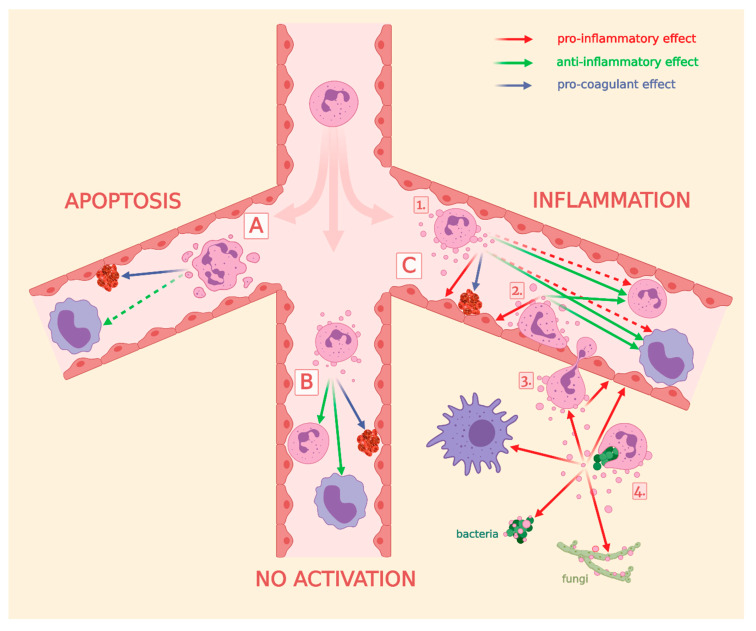
Overview of the role of PMN-EVs in intercellular communication, in coagulation and in pathogen elimination. Part (**A**) effects of apoEVs, (**B**) effects of spontaneous produced EVs, and (**C**) effects of EVs generated during inflammation. Numbers represent certain steps of PMN activation: (**1**) circulating PMNs, (**2**) endothelium-attached PMNs, (**3**) PMNs during extravasation, (**4**) PMNs after phagocytosing pathogens. Green arrows represent anti-inflammatory, red arrows represent pro-inflammatory, blue arrows represent pro-coagulant effects. Broken arrows represent non-consensual effects. References behind the arrows are listed in Table 2.

**Table 1 cells-09-02718-t001:** Overview of publications studying the biological function of neutrophil-derived EVs (extracellular vesicles).

	PMN-EV Induction Stimulus	Target	Effect	PMN Purity	PMN Viability	EV Isolation Method	EV Characterization Method	EV Diameter [nm]	EV Storage	Ref.
Unstimulated	Spontaneous release	HMDM	Bacterial killing ↓	? (nuclear morphology analyzed with light microscopy)	95% (Trypan Blue)	DC	FC, EM	50–300	−80 °C	[85]
		PMN, HUVEC, plasma	Anti-inflammatory, PMN ROS production ↓, pro-coagulant	>95%	?	DC + F	FC, DLS, NTA, EM	80–1000	none	[86]
	Apoptosis induction	none	No pro-inflammatory effect	?	n/a	no isolation	FC	?	?	[87]
		Monocytes, HMDM	Mostly anti-inflammatory, but IL-10 production of HMDM ↓	>90% (CD15 FC)	n/a	no isolation	FC	?	−70 °C	[88]
		PMN	ROS production ↓, Leishmania killing ↓	>99.9% (Diff Quik)	n/a	no isolation	FC	?	n/a	[89]
		Th cells	Anti-inflammatory	?	n/a	DC + F	FC, NTA	100–400	?	[90]
		HMDM	Anti-inflammatory	?	n/a	no isolation	FC	?	?	[91]
		PMN, HUVEC, plasma	PMN ROS production delayed, pro-coagulant	>95%	?	DC + F	FC, DLS, NTA, EM	80–1000	none	[86]
Bacterial byproducts	fMLP	HMDM	Anti-inflammatory	?	?	DC + F	?	?	−80 °C	[92]
		MoDC	Anti-inflammatory, anti-phagocytic	?	?	DC + F	FC	?	−80 °C	[93]
		HMDM	Anti-inflammatory	?	?	DC + F	none	?	−80 °C	[94]
		Peritoneal macrophages	Anti-inflammatory	?	?	DC	FC, EM	50–500	?	[95]
		NK cells	Anti-inflammatory	95% or 99% (FC)	?	DC	FC	200–1000	?	[96]
		PMN, systemic	PMN recruitment ↓, PMN-EC interaction ↓	?	?	DC	FC	?	?	[97]
		HUVEC	Pro-inflammatory, TF expression ↑	?	?	F or DC	FC	?	?	[98]
		HUVEC	Pro-inflammatory	?	?	DC + F	FC	?	?	[99]
		Human coronary endothelial cells	Pro-inflammatory, pro-migratory	?	?	DC + dialysis	FC, TRPS—human, NTA—mouse	280 (human), 165 (mouse)	?	[100]
		BMEC	Vascular permeability ↑	?	?	DC	FC, NTA	100–300	?	[101]
		IEC	Delivers pro-inflammatory miR content, genomic instability, impaired wound healing	Human: ?, Mouse: 85–90%	?	DC	EM	?	?	[102]
		PLT	Arachidonic acid transfer to PLT, causing TXA2 release and subsequent pro-inflammatory EC activation	?	?	ExoQuick-TC kit	?	?	?	[103]
		HUVEC	Non-adherent PMN-derived EVs: anti-inflammatory, vasoprotective. Adherent PMN-derived EVs: pro-inflammatory, vasoreactive	?	?	DC	FC	?	−80 °C	[104]
		HMDM	Pro-inflammatory, bacterial killing ↑	>98%	>98%	DC	FC, NTA, EM	100–200	4 °C <24 h	[105]
		HMDM, PMN, systemic	Pro- and anti-inflammatory, bacterial killing ↑, PMN and macrophage ROS production ↑	?	?	DC	FC	2000–3000	−80 °C	[106]
		ECM	Neutrophil elastase-dependent degradation of ECM	?	>95% (Trypan Blue)	DC	FC, NTA, EM	100	−80 °C or fresh	[107]
		IEC	Disruption of epithelial intercellular adhesion, enhanced transepithelial migration	Human: ?, Mouse: 85–90%	?	DC	FC, EM	100–800	?	[108]
		Vascular permeability	Maintaining the integrity of the microvascular barrier	?	?	no isolation	FC	?	?	[109]
		*S. aureus*	Binding to opsonized bacteria	>98%	>99% before and after stimulation (Trypan Blue)	DC + F	EM	?	?	[110]
	fMLP or fMLP + LatrB	IEC	Inhibition of epithelial wound healing via MPO delivery	Human: ?, Mouse: 85–90%	?	no isolation	FC, EM	600	?	[111]
	GM-CSF + (?) fMLP	PMN	Pro-inflammatory	?	?	DC + F	FC, EM	50–120 (purified from 50–500)	?	[112]
	fMLP + LPS	PMN, HMDM	ROS production ↑	?	?	DC	FC	?	?	[113]
	LPS	P1EC, artery rings	Pro-inflammatory, oxidative stress ↑, TF expression ↑	n/a (splenocytes)	?	DC	TRPS	200–500	?	[114]
		Airway smooth muscle cells	Proliferation	99.5% (Cytospin slide + Protocol Hema 3 staining)	97.75% (ADAM cell counter)	Size-exclusion chromatography	DLS, EM	30–80	−80 °C	[115]
		PLT	Platelet activation and co-aggregation with PMN, delivery of PAF receptor	?	?	DC	none	?	?	[116]
		PLT	Platelet activation	?	?	DC + F	FC	<1000	−80 °C	[117]
Endogenous pro-inflammatory mediators	TNFα	HDMD, joints, macrophage-FLS co-culture system	Anti-inflammatory	?	?	DC	FC, NTA	70–400	?	[118]
		IEC	Delivers pro-inflammatory miR content, genomic instability, impaired wound healing	Human: ?, Mouse: 85–90%	?	DC	EM	?	?	[108]
		Embryonic kidney cells	Transfer of kinin B1-receptors, calcium influx	?	?	DC	FC, EM	150	−80 °C	[119]
	IFN-γ	IEC	Delivers pro-inflammatory miR content, genomic instability, impaired wound healing	Human: ?, Mouse: 85–90%	?	DC	EM	?	?	[108]
		PMN, HUVEC	Mainly pro-inflammatory and pro-migratory, but reduced increase in EC permeability upon LPS treatment	n/a (stimulation in whole blood)	n/a (stimulation in whole blood)	? (DC)	FC	?	?	[120]
	GM-CSF	PMN, HUVEC	Mainly pro-inflammatory and pro-migratory, EC ROS production ↑, but reduced increase in EC permeability upon LPS treatment	n/a (stimulation in whole blood)	n/a (stimulation in whole blood)	? (DC)	FC	?	?	[120]
	C5a	HMDM	Anti-inflammatory	?	?	DC + F	none	?	−80 °C	[94]
		NK	Anti-inflammatory	95% or 99% (FC)	?	DC	FC	200–1000	?	[96]
		PMN, whole blood	Pro-inflammatory, ROS production ↑, MPO release ↑	?	?	DC	FC	300–1000	−80 °C	[121]
	PAF	PMN, systemic	PMN recruitment ↓, PMN-EC interaction ↓	?	?	DC	FC	?	?	[97]
		PLT	Platelet activation	?	?	DC + F	FC	<1000	−80 °C	[117]
	IL-8	NK	Anti-inflammatory	95% or 99% (FC)	?	DC	FC	200–1000	?	[96]
	CXCL-2	Vascular permeability	Maintaining the integrity of the microvascular barrier	?	?	no isolation	FC	?	?	[109]
Pathogens	*M. tuberculosis*	HMDM	Bacterial killing ↓	?, but nuclear morphology analyzed with light microscopy	95% (Trypan Blue)	DC	FC, EM	50–300	−80 °C	[85]
	*M. tuberculosis*	HMDM	Pro-inflammatory, ROS production ↑, autophagy ↑, bacterial killing ↑	>98%	>98%	DC	FC, NTA, EM	100–700	4 °C <24h	[105]
	Ops. *A. fumigatus*	*A. fumigatus*	Antifungal effect	>95%	>98%	DC + F	FC, NTA, EM	?	−80 °C or fresh	[122]
	*P. aeruginosa*	*P. aeruginosa*	Antibacterial effect	?	?	no isolation	none	?	?	[123]
	Ops. *S. aureus*	Ops. and non-ops. *S. aureus*, *E. coli*	Binding to bacteria, antibacterial effect	>95%	80–85% (EB)	DC + F	FC, DLS, EM	100, 200–800	?	[124]
	Ops. zymosan	*S. aureus*, *E. coli*	Antibacterial effect	>95%	?	DC + F	FC	?	?	[125]
		PMN, HUVEC, plasma	Pro-inflammatory, PMN ROS production ↑	>95%	?	DC + F	FC, DLS, NTA, EM	80–1000	none	[86]
Pharmacological stimuli	PMA	MoDC	Anti-inflammatory, Th2 polarization	?	?	DC	FC, DLS	50–600	−80 °C	[126]
		HMDM	Pro-inflammatory	>98%	>98%	DC	FC, NTA, EM	100–300	4 °C <24 h	[105]
		IEC	Inhibition of epithelial wound healing via MPO delivery	Human: ?, Mouse: 85–90%	?	DC	FC, EM	600	?	[111]
		*S. aureus*	Binding to opsonized bacteria	>98%	>99% before and after stimulation (Trypan Blue)	DC + F	EM	?	?	[110]
		Plasma, NET	Pro-coagulant (intrinsic), NET-binding	?	?	no isolation	FC, EM	?	?	[127]
		PLT	Platelet activation	?	?	DC + F	FC	<1000	−80 °C	[117]
	PMA + A23187	P1EC, artery rings	Pro-inflammatory, oxidative stress ↑, TF expression ↑	n/a (splenocytes)	?	DC	TRPS	200–500	?	[114]
	A23187	HUVEC	MPO-mediated cytotoxicity	>90% (FC CD66b)	?	DC	FC, EM	<1000	4 °C	[128]
	Ionomycin	*S. aureus*	Binding to opsonized bacteria	>98%	>99% before and after stimulation (Trypan Blue)	DC + F	EM	?	?	[110]
	L-NAME	PMN	Pro-migratory	>97% (hemocytometer)	>95% (Trypan Blue)	DC	FC, EM	?	?	[129]
Pathophysiological environment	Sepsis	THP-1	Pro-inflammatory, pro-phagocytic	n/a (peritoneal and BAL EVs)	n/a (peritoneal and BAL EVs)	C	FC	300–1100	?	[130]
		HUVEC, Plasma, ops. *S. aureus*	Pro-inflammatory, pro-coagulant, binding to ops. bacteria	>95% (FC)	?	DC	FC, NTA	50–800	?	[131]
		Ops. and non-ops. *S. aureus*, *E. coli*	Binding to bacteria	n/a (plasma EVs)	n/a (plasma EVs)	DC + F	FC	?	?	[124]
	Sepsis + LL37	*E. coli*	Antibacterial effect	90% (Giemsa)	?	DC	FC	500–1000	−80 °C	[132]
	Sepsis + thioglycolate i.p.	Peritoneal macrophages, systemic	Pro- and anti-inflammatory, bacterial clearance ↓, mortality ↑	n/a (peritoneal EVs)	n/a (peritoneal EVs)	DC	FC	?	?	[133]
	Cystic fibrosis/primary ciliary dyskinesia	Airways	Pro-inflammatory	n/a (sputum EVs)	n/a (sputum EVs)	DC	FC	?	4 °C	[134]
	Pancreatitis	Pancreas acinar cells, systemic	Pro-inflammatory, tissue injury ↑	n/a (pancreatic EVs)	n/a (pancreatic EVs)	DC	EM	?	n/a (pancreatic EVs)	[135]
	ANCA vasculitis	none	Pro-coagulant (extrinsic)	?	?	DC	FC	?	?	[136]
	TNFα + ANCA	HUVEC	Pro-inflammatory, pro-coagulant, ROS production ↑	?	?	DC	FC	?	Frozen (no temp. data)	[137]
	Rheumatoid arthritis + TNFα	HDMD, joints, macrophage-FLS co-culture system	Anti-inflammatory	?	?	DC	FC, NTA	70–400	?	[118]
	MSU i.p.	Peritoneal macrophages	Anti-inflammatory	n/a (peritoneal EVs)	n/a (peritoneal EVs)	DC	FC, EM	50–500	?	[95]
	Gout	Peritoneal macrophages	Anti-inflammatory	n/a (synovial EVs)	n/a (synovial EVs)	DC	FC, EM	50	?	[95]
	acLDL	Human coronary endothelial cells	Pro-inflammatory, pro-migratory	?	?	DC + dialysis	FC, TRPS—human, NTA—mouse	280 (human), 165 (mouse)	?	[100]
	Hyperglycemia	None	Release of EVs carrying IL-1β	?	>78% after EV isolation (Trypan Blue)	no isolation	FC	300–1000	?	[138]

Abbreviations of Table 1. ‘?’: not communicated; n/a: not applicable; LatrB: Latrunculin B; Ops.: opsonized; MSU: monosodium urate; i.p.: intraperitoneal; HMDM: human monocyte derived macrophage; HUVEC: human umbilical vein endothelial cell; MoDC: monocyte derived dendritic cell; NK: natural killer; BMEC: brain microvascular endothelial cell; IEC: intestinal epithelial cell; PLT: platelet; ECM: extracellular matrix; P1EC: primary porcine endothelial cell; FLS: fibroblast-like synoviocyte; ROS: reactive oxygen species; EC: endothelial cell; MPO: myeloperoxidase; FC: flow cytometry; EM: electron microscopy; DLS: dynamic light scattering; NTA: nanoparticle tracking analysis; TRPS: tunable resistive pulse sensing; DC: differential centrifugation; F: filtration.

**Table 2 cells-09-02718-t002:** List of publications behind Figure 4 and impact of PMN-EVs on target cells.

Group (Figure 4)		Effect	PMN-EV Induction Stimulus	Target	References
A		No effect	apoptosis	PMN	[86]
					[89]
				Monocyte/macrophage	[87]
					[91]
				T-cell	[90]
		Anti-inflammatory		Monocyte/macrophage	[88]
		Pro-coagulant		Coagulation	[86]
B		Anti-inflammatory	spontaneous release	Monocyte/macrophage	[85]
				PMN	[86]
		Pro-coagulant		Coagulation	[86]
C	1	Anti-inflammatory	fMLP	Monocyte/macrophage	[92]
					[95]
					[93]
					[104]
					[94]
			C5a		[94]
			TNFα		[118]
			fMLP, IL-8	NK	[96]
			PAF	PMN	[97]
		Pro-inflammatory	fMLP	Endothelium/HUVEC	[99]
					[98]
					[100]
					[101]
			LPS		[114]
			TNFα		[137]
			TNFa, GM-CSF, IFN-γ		[102]
			fMLP + GM-CSF	PMN	[112]
			C5a		[121]
			fMLP+LPS	Phagocytes	[113]
		Pro-coagulant	fMLP	Coagulation	[98]
			TNFα + ANCA		[137]
			LPS		[117]
					[116]
	2	Anti-inflammatory	fMLP	Monocyte/macrophage	[97]
		Pro-inflammatory		Phagocytes	[106]
				Endothelium/HUVEC	[104]
					[103]
	3	Pro-inflammatory	fMLP	Endothelium/HUVEC	[108]
					[111]
					[102]
	4	Pro-inflammatory	ops. zymosan	PMN	[86]
				Bacteria	[149]
			ops. bacteria	Bacteria	[124]
			ops. fungi	Fungi	[122]
			*M. tuberculosis*	Monocyte/macrophage	[105]
		Anti-inflammatory			[85]

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
