# Peer review of "The Functional Heterogeneity of Neutrophil-Derived Extracellular Vesicles Reflects the Status of the Parent Cell"

_cells, 2020, doi:10.3390/cells9122718_

Round 1
Reviewer 1 Report
In this manuscript, the authors discuss neutrophil-derived extracellular vesicles (EVs), and the functional heterogeneity of these EVs under different conditions of their secreting cells.
The review is interesting, and contributes the important content related to emerging interest of researchers in studying neutrophil-derived extracellular vesicles (EVs). However, this reviewer observes an important missing piece of information that many readers may want to know. Especially those struggling with isolating EVs from neutrophils.
My comments and suggestions are appended below;
(1). Although, the authors have listed some EV isolation techniques in table 1. Nevertheless, it is important to consider that related to the topic under discussion, several researchers still find difficulties in isolating EVs from non-proliferating neutrophil (PMN) and those obtained from Buffy coats of healthy donors, as well as from patient blood and synovial fluids using different methods. Please include a dedicated section for the isolation and purification of EVs from Neutrophils.
Here in this section, I suggest authors to discuss following three aspects;
- the starting material for Neutrophil-EVs (e.g, Buffy coats of healthy donors, patients’ peripheral blood, PBMCs, synovial fluids under pathological conditions, as well as cell culture conditioned media.
- Brief protocols (and if there are advantages and disadvantages of each method for isolating neutrophil-EVs). As several researchers still find difficulties in isolating EVs from non-proliferating neutrophils/PMN and those obtained from Buffy coats of healthy donors, as well as from patient blood and synovial fluids.
- Recommendations to overcome those hurdles in isolation and purification.
All these 3 aspects here needed to be related to neutrophils/PMN/macrophages.
Other minor comments:
(2). Literature needs to be updated, there are several recent studies about isolation, characterization, and functional studies of extracellular vesicles from neutrophils/PMN, which can be included in this discussion.
(3). An addition of a schematic would help as a quick guide to demonstrate the effect of PMN-EVs released without stimulation and effects of EVs upon release against endogenous activation, as well as their non-cellular effects.
Some corrections in presentation of text:
- In the abstract: authors refer that ‘’We propose’’. Based on?
- In the abstract: authors refer that ‘’We identified Mac1’’. How, which method authors have used?. Rather it was identified by other authors.
- PMN – write at full, when first time mentioned in the text, and later use the abbreviation.
Author Response
Answer to R1:
Thank you for the comments and suggestions. Please find our answers below your questions.
In this manuscript, the authors discuss neutrophil-derived extracellular vesicles (EVs), and the functional heterogeneity of these EVs under different conditions of their secreting cells.
The review is interesting, and contributes the important content related to emerging interest of researchers in studying neutrophil-derived extracellular vesicles (EVs). However, this reviewer observes an important missing piece of information that many readers may want to know. Especially those struggling with isolating EVs from neutrophils.
My comments and suggestions are appended below;
(1). Although, the authors have listed some EV isolation techniques in table 1. Nevertheless, it is important to consider that related to the topic under discussion, several researchers still find difficulties in isolating EVs from non-proliferating neutrophil (PMN) and those obtained from Buffy coats of healthy donors, as well as from patient blood and synovial fluids using different methods. Please include a dedicated section for the isolation and purification of EVs from Neutrophils.
Here in this section, I suggest authors to discuss following three aspects;
the starting material for Neutrophil-EVs (e.g, Buffy coats of healthy donors, patients’ peripheral blood, PBMCs, synovial fluids under pathological conditions, as well as cell culture conditioned media.
Brief protocols (and if there are advantages and disadvantages of each method for isolating neutrophil-EVs). As several researchers still find difficulties in isolating EVs from non-proliferating neutrophils/PMN and those obtained from Buffy coats of healthy donors, as well as from patient blood and synovial fluids.
Recommendations to overcome those hurdles in isolation and purification.
All these 3 aspects here needed to be related to neutrophils/PMN/macrophages.
We accepted the suggestion of the reviewer and added new paragraphs on PMN-EV isolation and gave recommendations on PMN-EV preparation and handling (lines 187-215). EVs originating from PBMCs and macrophages do not belong to the focus of this review, as it focusses on neutrophil derived EVs.
Other minor comments:
(2). Literature needs to be updated, there are several recent studies about isolation, characterization, and functional studies of extracellular vesicles from neutrophils/PMN, which can be included in this discussion.
We did our best to collect and review all publications on PMN that meet the minimum criteria of EV research (MISEV criteria), discuss functional data on PMN-EVs or work with pure PMN-EV isolates. There are many other excellent studies that are not specific for PMNs or present descriptive data on PMN derived EVs. Although these works show important data on PMN-EVs, they do not fit in our current aim to overview the functional heterogeneity of PMN-EVs.
Yet, it is possible that relevant articles are missing from our manuscript. We would welcome any specific suggestions from the reviewer to complete this review.
(3). An addition of a schematic would help as a quick guide to demonstrate the effect of PMN-EVs released without stimulation and effects of EVs upon release against endogenous activation, as well as their non-cellular effects.
Figure 4 and the belonging table 2 show and list the effects of PMN-EV released without stimulation (part B) and upon endogenous activation (part C 1, 2, 3). We completed part B with their anti-inflammatory effect on monocytes/macrophages. Non-cellular effects are also demonstrated in Fig.4 and table 2.
Some corrections in presentation of text:
In the abstract: authors refer that ‘’We propose’’. Based on?
We have rephrased the abstract to make it clear, what inspired us (line 23).
In the abstract: authors refer that ‘’We identified Mac1’’. How, which method authors have used?. Rather it was identified by other authors.
That sentence is replaced and rephrased. (The work that describes the role of Mac1 in pro-inflammatory EV generation was published by the authors of this review.) (lines 18-20)
PMN – write at full, when first time mentioned in the text, and later use the abbreviation.
The abbreviation PMN is used the first time in lines 63-64 and at that place it is written at full.
Reviewer 2 Report
Remarks to the authors:
The manuscript by Kolonics and colleagues describe that neutrophils are capable of producing different types of EVs that modulate different mechanism in the immune response, depending of the neutrophiles activation. The EVs could have a similar role as the pro-inflammatory and anti-inflammatory cytokines, altering the environmental system.
Major Comments:
Here the authors highlight a new concept about of the role of tumor-derived EVs. It could have a similar role as the pro-inflammatory and anti-inflammatory cytokines depending on the neutrophil’s stimulation could alter the environmental system.
The authors explain about the exosome and apoptotic body formation but didn’t introduced a satisfactory explanation about the exosome formation.
For Example: Exosomes are small EVs (sEVs). sEVs are formed intracellularly by inward budding of the endosomal membrane resulting in sequestration of RNA, DNA, proteins, and lipids into intraluminal vesicles (ILVs) within the lumen of multivesicular bodies (MVBs).……. the plasma membrane leads to release of ILVs which are then termed sEVs; this budding event during sEV formation occurs in a reverse membrane orientation
Could explain the “gold standard method” for the recognizing of EV’s like CD63, Alix, CD81 markers, electric microscopy and nano sight?
Tumor-derived EVs not only interact with T-Cells , also can interact with lymphocytes binding to cellular MHC receptors through ligands or antigens exposed on their membrane or carried by them, thus altering immune function. In addition, phagocytic cells, such as macrophages and dendritic cells.
Could you add these information?
Minor Points:
Similar to other cell types, neutrophilic granulocytes also release extracellular vesicles (EVs), mainly medium-sized microvesicles/microparticles. Published data on the physical parameters (size, density) and chemical composition (surface proteins, proteomics) of neutrophil-derived EVs show” reasonable” agreement.
Its more accurate to say that the authors reach a consensus about the physical parameters and chemical compositions previously commented, more than a “Reasonable Agreement”.
Author Response
Answers to R2:
Thank you for the comments and suggestions. Please find our answers below your questions.
Remarks to the authors:
The manuscript by Kolonics and colleagues describe that neutrophils are capable of producing different types of EVs that modulate different mechanism in the immune response, depending of the neutrophiles activation. The EVs could have a similar role as the pro-inflammatory and anti-inflammatory cytokines, altering the environmental system.
Major Comments:
Here the authors highlight a new concept about of the role of tumor-derived EVs. It could have a similar role as the pro-inflammatory and anti-inflammatory cytokines depending on the neutrophil’s stimulation could alter the environmental system.
The authors explain about the exosome and apoptotic body formation but didn’t introduced a satisfactory explanation about the exosome formation.
For Example: Exosomes are small EVs (sEVs). sEVs are formed intracellularly by inward budding of the endosomal membrane resulting in sequestration of RNA, DNA, proteins, and lipids into intraluminal vesicles (ILVs) within the lumen of multivesicular bodies (MVBs).……. the plasma membrane leads to release of ILVs which are then termed sEVs; this budding event during sEV formation occurs in a reverse membrane orientation
Could explain the “gold standard method” for the recognizing of EV’s like CD63, Alix, CD81 markers, electric microscopy and nano sight?
We agree with the reviewer that the formation of extracellular vesicles and the investigation of EVs are very important issues. However, we were invited to summarize the functional heterogeneity of neutrophil derived EVs, that is why we turned to functional aspects of EVs after a short introduction. We feel that a satisfyingly detailed section on EV formation and EV examination is beyond the scope of this review. Fortunately there are many reviews focusing on different EV types and on their characteristics, isolation and investigation in general. We completed the text with these references (lines 58-61).
Tumor-derived EVs not only interact with T-Cells , also can interact with lymphocytes binding to cellular MHC receptors through ligands or antigens exposed on their membrane or carried by them, thus altering immune function. In addition, phagocytic cells, such as macrophages and dendritic cells.
Could you add these information?
We have clarified the relevant part of the text and added important references that discuss this field in details (lines 73-75).
Minor Points:
Similar to other cell types, neutrophilic granulocytes also release extracellular vesicles (EVs), mainly medium-sized microvesicles/microparticles. Published data on the physical parameters (size, density) and chemical composition (surface proteins, proteomics) of neutrophil-derived EVs show” reasonable” agreement.
Its more accurate to say that the authors reach a consensus about the physical parameters and chemical compositions previously commented, more than a “Reasonable Agreement”.
We rephrased the sentence accordingly. (line 11)
Reviewer 3 Report
In the manuscript by Ferenc Kolonics et al., authors summarize the state of art of neutrophilic granulocyte derived EVs considering a variety of aspects.
The topic of this work is interesting, well written in the English language, and also well supported by tables and figures. Nevertheless, the review needs substantial and drastic adjustments (see the specific comments) and a logical sequence is missing. The introduction is disconnected from chapter 2 and the conclusion part is extremely long and not incisive, moreover its structure is similar to a discussion part as it contains new concepts (figures and tables included) that should be in the center of the review. Another general consideration is that I don’t see the scientific contribution of this review, therefore I would suggest expanding the discussion of some sections (see specific comments) in order to make it more impactful.
I have the following specific comments:
- Abstract
line 23-25: This sentence written in this way results disconnetted, If you want to leave it I would at least encourage the writer to rearrange it. - Introduction:
This paragraph is too general and in this regard I would like to encourage the writer to focus more on the main purpose of this review, concerning the EV of neutrophils.
Line 40: “tubular EV” as far as I know, are poorly known and discussed, thus if mentioned it would be interesting known what is the biological role, what is their size, where are they synthesized? By which type of cells? Etc..
- EVs in intercellular communication:
- Line 78: “EV are used for immune modulation” also EV derived from mesenchymal stem cells are immunomodulators (see: Bari et al Freeze-dried and GMP-compliant pharmaceuticals containing exosomes for acellular mesenchymal stromal cell immunomodulant therapy. Nanomedicine (Lond). 2019 Mar;14(6):753-765. doi: 10.2217/nnm-2018-0240. Epub 2019 Feb 11. PMID: 30741596)
- Non-cellular effects of EVs:
In my opinion it is a paragraph that would be worth investigating a little more. There are only few informations and regards only the biological processes. Line 84-85:
“EV characterization identified biological processes that are related directly to EVs”,
Are these processes referable only to EVs or parent cells also? must be specified.
- EVs in pathological conditions:
this paragraph could be better rearrange, it should be better explain and/or clarified the pathological role of the different EVs classified based on their "mother" cells (tumor cells or other type). Line 99: “Now we know that EV..”,
which EVs? those produced by cancer cells? you need to specify which EV you are referring to. Same thing for line 107 which EVs are you referring to? Produced by which cells?
- Diagnostics and therapy:
the part related to the therapeutic use of EVs is too general. If you want to leave this section it should be expanded and be more argued (especially the drug delivery part). Moreover which EVs are used for this purpose?
(J Control Release. 2017 Sep 28;262:104-117. doi: 10.1016/j.jconrel.2017.07.023. Epub 2017 Jul 20. PMID: 28736264). Line 132-133: this concept could also be expanded because EVs from MSCs also have immunomodulatory effects.
- Neutrophil derived EVs:
Table 1 needs to be discussed in detail in the text, otherwise it is a dry list. Moreover there are many question marks, perhaps this aspect should also be commented inside the text. Line 188: evaluation of the number of EVs is not the only approach. Many authors for example estimate EVs production in terms of total protein and total phospholipids content. This aspect must be addressed. Line 190-191: The source of the data are not shown, since it is a this is a review references should be addressed. Figure 2 line 197: this description should be under the figure. Line 213-214-215: Here, should be better investigate what is the conclusion of the proteomics studies? what do they lead to? Line 224: “inappropriate storing conditions affect the function of EVs” Again, this is a very big problem which in my opinion is argued superficially. what are the conservation methods proposed? freezing and freeze-drying? what are the limits and advantages of these 2 different approaches? do we need to use cryopreservatives? If so, which one?
- Conclusions:
As mentioned above, conclusions are too long and not very incisive. Figures and tables should be in the body/central part of the text not in the end.
- References: Please double check the references order, i.e. line 308: Peretti et al is not ref (73)
Author Response
Answers to R3:
Thank you for the comments and suggestions. Please find our answers below your questions.
In the manuscript by Ferenc Kolonics et al., authors summarize the state of art of neutrophilic granulocyte derived EVs considering a variety of aspects.
The topic of this work is interesting, well written in the English language, and also well supported by tables and figures. Nevertheless, the review needs substantial and drastic adjustments (see the specific comments) and a logical sequence is missing. The introduction is disconnected from chapter 2 and the conclusion part is extremely long and not incisive, moreover its structure is similar to a discussion part as it contains new concepts (figures and tables included) that should be in the center of the review. Another general consideration is that I don’t see the scientific contribution of this review, therefore I would suggest expanding the discussion of some sections (see specific comments) in order to make it more impactful.
New paragraphs are added to highlight the scientific contribution of this review. A new section is added to link introduction better to the central part of the review (lines 153-161). And accepting the opinion of the reviewer, we renamed ‘conclusion’ to ‘discussion’, as the third part of our manuscript is more like a discussion than a conclusion.
I have the following specific comments:
Abstract
line 23-25: This sentence written in this way results disconnetted, If you want to leave it I would at least encourage the writer to rearrange it.
Sentence is replaced and rephrased. (lines 18-20)
Introduction:
This paragraph is too general and in this regard I would like to encourage the writer to focus more on the main purpose of this review, concerning the EV of neutrophils.
We agree with the Reviewer that a focused introduction serves better, unfortunately EV research is young and the basics can not be explained enough times. We think it is important to clarify what EVs are and what kind of EV subgroups exist to understand the following part of the manuscript.
Line 40: “tubular EV” as far as I know, are poorly known and discussed, thus if mentioned it would be interesting known what is the biological role, what is their size, where are they synthesized? By which type of cells? Etc..
According to the above mentioned postulates, we rephrased the sentence to skip “tubular EVs”. (lines 42)
EVs in intercellular communication:
Line 78: “EV are used for immune modulation” also EV derived from mesenchymal stem cells are immunomodulators (see: Bari et al Freeze-dried and GMP-compliant pharmaceuticals containing exosomes for acellular mesenchymal stromal cell immunomodulant therapy. Nanomedicine (Lond). 2019 Mar;14(6):753-765. doi: 10.2217/nnm-2018-0240. Epub 2019 Feb 11. PMID: 30741596)
The text is completed with the MSC derived EVs. (line 85)
Non-cellular effects of EVs:
In my opinion it is a paragraph that would be worth investigating a little more. There are only few informations and regards only the biological processes. Line 84-85:
“EV characterization identified biological processes that are related directly to EVs”,
Are these processes referable only to EVs or parent cells also? must be specified.
The definition of non-cellular effects is now explained more in details. (lines 92-94)
EVs in pathological conditions:
this paragraph could be better rearrange, it should be better explain and/or clarified the pathological role of the different EVs classified based on their "mother" cells (tumor cells or other type). Line 99: “Now we know that EV..”,
which EVs? those produced by cancer cells? you need to specify which EV you are referring to. Same thing for line 107 which EVs are you referring to? Produced by which cells?
We have completed the text with information on mother cells (lines 108-111). In line 116 the sentence: “EVs play a role in immunologic processes as well.” refers to EVs in general. This statement is explained with examples in the following sentences. In line with ‘focused introduction’ (chapter 1), our goal was to demonstrate the functional diversity of EV in general instead of explaining all phenomena in detail. This presented functional diversity of PMN-EVs was compared to the functional diversity of EVs in general demonstrated in chapter 1.
Diagnostics and therapy:
the part related to the therapeutic use of EVs is too general. If you want to leave this section it should be expanded and be more argued (especially the drug delivery part). Moreover which EVs are used for this purpose? (J Control Release. 2017 Sep 28;262:104-117. doi: 10.1016/j.jconrel.2017.07.023. Epub 2017 Jul 20. PMID: 28736264).
In the first part of our review we present the importance of EVs in general, as EVs and EV-research do not belong to the widely known fields. We think it is important to show, that there are efforts to use EVs in the therapy. As an example we refer MSC derived EVs as the most investigated EVs for clinical use. We also refer EVs as drug delivery system option. We reformatted this section to make it more clear, and inserted the publication suggested by the reviewer to give options for the reader to find detailed information on these areas. (lines 142-146)Line 132-133: this concept could also be expanded because EVs from MSCs also have immunomodulatory effects.
We have completed the text with the immunomodulatory effects of MSC EV in the revised version. ( line 85)Neutrophil derived EVs:
Table 1 needs to be discussed in detail in the text, otherwise it is a dry list. Moreover there are many question marks, perhaps this aspect should also be commented inside the text.
We added a new section that is based on the table 1 and explains the meaning and consequences of question marks. (187-204)
Line 188: evaluation of the number of EVs is not the only approach. Many authors for example estimate EVs production in terms of total protein and total phospholipids content. This aspect must be addressed.
Text is supplemented with bulk measurements. (lines 237-241)
Line 190-191: The source of the data are not shown, since it is a this is a review references should be addressed.
References are transferred from figure legends into the text. (line 244)
Figure 2 line 197: this description should be under the figure.
It is replaced accordingly. (line 250)
Line 213-214-215: Here, should be better investigate what is the conclusion of the proteomics studies? what do they lead to?
In this review we want to focus on the functional heterogeneity of PMN-derived EVs. Detailed analysis of the proteomics studies are given in the original papers to which we refer.
Line 224: “inappropriate storing conditions affect the function of EVs” Again, this is a very big problem which in my opinion is argued superficially. what are the conservation methods proposed? freezing and freeze-drying? what are the limits and advantages of these 2 different approaches? do we need to use cryopreservatives? If so, which one?
We have added a new section that gives suggestions on storing of PMN-EVs based on our previous publication (ref 94). As we know, there are no other publications on storing of PMN-EVs. Findings on storage of other EVs can not be applied to PMN-EVs without caution, as PMN-EVs are filled with antibacterial enzymes that can autodigest vesicular structures as well. (line 213)
Conclusions:
As mentioned above, conclusions are too long and not very incisive. Figures and tables should be in the body/central part of the text not in the end.
We renamed ‘conclusion’ to ‘discussion’ as the third part of our manuscript is more like a discussion than a conclusion.
References: Please double check the references order, i.e. line 308: Peretti et al is not ref (73)
We have completed and corrected the list of references.
Round 2
Reviewer 1 Report
The authors have addressed the comments and have further improved their manuscript.
However, one of the comments still needs to be updated.
The following statement in the abstract. We also identified Mac-1 integrin as key factor that switches....
As highlighted in previous comments, please note that Mac-1 integrin was not identified in this study. This reviewers understands that this was identified by this research group, but in previous study. What we write here, or reviewers suggests is for readers perspective. Please simply update the statement like, previously we identified Mac-1 integrin as key factor that switches....
or previously we have shown that Mac-1 integrin........
Authors may also remove the word ''we indicate'' from statement mentioned below. (generally such statement ''we indicate'' or ''our data indicates'' is used for original research work, however current manuscript is a review article and is based on previously indicated publications).
Finally, based on the observed functional heterogeneity of differently triggered EVs we indicate that neutrophils are capable of.....
This can be rephrased as;
Finally, the functional heterogeneity of differently triggered EVs indicates that neutrophils are capable of.....
I have no further comments and I already endorse the publication of this manuscript after improving the presentation of text for the readers.
Author Response
We accepted the suggestions of Reviewer1 and modified the relevant sentences.
Reviewer 3 Report
I endorse the publication of the paper
Author Response
Thank you for accepting our answers.